# Premature Macrophage Activation by Stored Red Blood Cell Transfusion Halts Liver Regeneration Post-Partial Hepatectomy in Rats

**DOI:** 10.3390/cells11213522

**Published:** 2022-11-07

**Authors:** Nathalie Abudi, Omri Duev, Tal Asraf, Simcha Blank, Idit Matot, Rinat Abramovitch

**Affiliations:** 1The Goldyne Savad Institute of Gene Therapy, Hadassah Medical Organization, Jerusalem 91120, Israel; 2The Wohl Institute for Translational Medicine, Hadassah Medical Organization, Jerusalem 91120, Israel; 3Faculty of Medicine, Hebrew University of Jerusalem, Jerusalem 91121, Israel; 4The Anesthesia, Pain, and Intensive Care Division, Tel Aviv Medical Center, Sackler School of Medicine, Tel Aviv University, Tel Aviv 64239, Israel

**Keywords:** hepatectomy, liver regeneration, activated liver macrophages, heme oxygenase-1

## Abstract

Liver resection is a common treatment for various conditions and often requires blood transfusions to compensate for operative blood loss. As partial hepatectomy (PHx) is frequently performed in patients with a pre-damaged liver, avoiding further injury is of paramount clinical importance. Our aim was to study the impact of red blood cell (RBC) resuscitation on liver regeneration. We assessed the impact of RBC storage time on liver regeneration following 50% PHx in rats and explored possible contributing molecular mechanisms using immunohistochemistry, RNA-Seq, and macrophage depletion. The liver was successfully regenerated after PHx when rats were transfused with fresh RBCs (F-RBCs). However, in rats resuscitated with stored RBCs (S-RBCs), the regeneration process was disrupted, as detected by delayed hepatocyte proliferation and lack of hypertrophy. The delayed regeneration was associated with elevated numbers of hemorrhage-activated liver macrophages (Mhem) secreting HO-1. Depletion of macrophages prior to PHx and transfusion improved the regeneration process. Gene expression profiling revealed alterations in numerous genes belonging to critical pathways, including cell cycle and DNA replication, and genes associated with immune cell activation, such as chemokine signaling and platelet activation and adhesion. Our results implicate activated macrophages in delayed liver regeneration following S-RBC transfusion via HO-1 and PAI-1 overexpression.

## 1. Introduction

The healthy liver has a remarkable ability to regenerate, permitting extensive resections as well as living-donor liver transplantations. Liver regeneration is an orchestrated process that allows a rapid return to the original liver-to-body weight ratio following damage or partial loss of the liver [1,2,3]. Recently, improved post-resection survival rates have been reported in the treatment of patients with liver lesions, as compared with non-surgical strategies [4,5,6,7]. Accordingly, radical surgical approaches have been implemented that result in smaller remnant liver volumes, thereby requiring more efficient liver regeneration [8]. Yet, an adverse event that is frequently encountered during major liver surgery is significant bleeding, which necessitates RBC transfusion [9]. The liver is known to be susceptible to injury in low-flow states associated with acute and massive bleeding [10]; liver injury could, in turn, affect liver regeneration. The need for RBC transfusions increases the possibility that patients will receive stored RBC (S-RBC). Human RBCs are typically stored for as long as 35–42 days. The maximum storage time of RBCs is based on two criteria: the lack of hemolysis at the end of the storage period and an RBC survival rate of at least 75% 24 h after transfusion [11]. These criteria do not consider additional parameters which may affect the efficacy of S-RBC transfusion. These parameters are referred to as the storage lesion, which indicates changes in RBCs that occur with storage time, such as membrane rigidity, slowed metabolism, acidosis, and decreased concentrations of DPG and ATP [12,13]. In addition, remnant white blood cells release cytokines that may activate an inflammatory response in the patient receiving the transfusion [14]. Concerns have previously been raised regarding the safety and efficacy of transfusing S-RBCs in certain scenarios. Several groups have reported a positive correlation between the transfusion of older RBCs and clinically adverse outcomes [15,16,17]. In a previous study, we showed that in an in vivo rat model, acute bleeding induced significant liver injury with poor liver perfusion and oxygenation, and, importantly, the extent of the damage depended on the resuscitation protocol [18,19]. Although, in a clinical setting, the problems that arise with the use of aged blood transfusions have not been confirmed, as regards liver surgeries, it was shown that allogenic transfusion of RBCs or blood components is associated with increased post-operative complications and morbidity [20,21]. In our previous studies, we characterized the kinetics of liver regeneration, perfusion, and oxygenation in rats following PHx using MRI [22,23]. We showed that controlled bleeding in rats during PHx has detrimental effects on liver regeneration: the process is slower and occurs mainly through hepatocyte hypertrophy rather than via the hyperplasia that is observed after PHx in the context of minimal blood loss [24].

Kupffer cells (KCs), the resident liver macrophages, represent approximately 35% of the non-parenchymal cells in normal liver. The heterogeneity of KCs has been identified in different studies. The phenotype of KCs is beyond the traditional dogma of M1–M2 macrophages. KCs contribute to the resolution of liver injury and restoration of tissue architecture. The underlying mechanism varies according to damage factors and pathology. KCs release TNF-α and IL-6, which prime hepatocyte proliferation in in vivo liver regeneration after PHx [25]. However, the role of KCs in liver regeneration is unclear since conflicting data have emerged from studies performed in KC-depleted mice and rats. Some studies showed that KC depletion prior to PHx enhances liver regeneration, whereas in other reports, the process of liver regeneration was delayed [26,27,28]. The aim of the current study was to determine the effect of RBC storage time on the ability of the liver to regenerate effectively. Following PHx, controlled bleeding, and resuscitation with S-RBCs, we found that the liver regeneration process in our rat model was delayed and attenuated due to a decrease in hepatocyte proliferation and a lack of hypertrophy compared to animals receiving fresh RBC (F-RBC) transfusions. We also present evidence that KCs play a deleterious role in liver regeneration following S-RBC transfusion. In particular, the Mhem subgroup of macrophages might participate in this process due to their increased expression of Heme oxygenase-1 (HO-1). Taken together, our data emphasize the clinical relevance of RBC storage time on the well-being of patients following PHx and blood transfusion.

## 2. Materials and Methods

### 2.1. Animal Experiments

All animal experiments were performed in accordance with the guidelines of the Institutional Animal Care and Use Committee (IACUC) of the Hebrew University (NIH approval number OPRR-A01-5011). Adult male Sprague-Dawley rats weighing 300 ± 20 gr were used for all experiments.

#### 2.1.1. RBC Collection and Storage

Approximately 10 mL of blood was collected from the heart of each rat and centrifuged at 1400 rpm for 10 min. The serum was discarded, and the remaining RBCs were stored in blood bags with CPDA-1 at 4 °C.

#### 2.1.2. PHx, Controlled Bleeding and Resuscitation Procedures

Rats were anesthetized with an intraperitoneal injection of 100 mg/kg ketamine and 5 mg/kg xylazine. The femoral arteries were cannulated for controlled bleeding (3 mL of blood was extracted at a rate of 1 mL/min) and RBC transfusion (3 mL of blood was injected at a rate of 1 mL/min). PHx was performed by resection of 50% of the total liver mass by removing the median and left lateral lobes, as previously described [22,23]. Rats were sacrificed one, two, four, and seven days post-PHx (n = 6 rats/time point). Liver tissue was snap-frozen in liquid nitrogen and stored at −80 °C for RNA extraction or fixed in formalin for histological evaluation. Blood samples were also collected for blood cell count and measurement of liver enzyme levels and serum cytokine levels. Liver enzyme levels in serum [alanine aminotransferase (ALT) and aspartate aminotransferase (AST)] were determined using a Reflotron analyzer (Roche, Mannheim, Germany). For measurement of serum prothrombin time, sodium citrate (0.105 M) was immediately added to the blood sample at a ratio of 1:10 (*v*/*v*), and values were recorded with a Beckman Coulter ACL 9000 Coagulation Analyzer according to the manufacturer’s instructions.

#### 2.1.3. Experimental Groups

(1) **Naïve** (n = 8): no PHx, no RBC transfusion; (2) **PHx** (n = 24): animals underwent 50% PHx; (3) **F-RBC transfusion** (n = 24): animals underwent controlled bleeding prior to 50% PHx and resuscitation with F-RBCs, 10 min post-PHx; (4) **S-RBC transfusion** (n = 24): animals underwent controlled bleeding prior to 50% PHx and administration of 7-day-old S-RBCs, 10 min post-PHx. It was previously shown that storage of rat RBCs for one week in CPDA-1 produces a storage lesion similar to human RBCs stored for four weeks [15].

#### 2.1.4. KCs Depletion and HO-1 Inhibitor Administration

Selective KC depletion was performed as described previously [26] on an additional 14 rats/group that were sacrificed on days one or two post-PHx (7 rats/time point). Briefly, 24 h prior to PHx, 1.5 mL of liposomes encapsulating the drug, dichloromethylene-diphosphonate (Cl_2_MDP; clodronate; **Cld**) (kindly provided by Prof. Chezy Bernholtz, Hebrew University, Israel) were injected I.V. into the tail vein. Control rats received I.V. administration of saline (0.9% NaCI) in an equal volume. The HO-1 inhibitor, **SnPP** (Sn(IV) mesoporphyrin IX dichloride; Frontier Scientific, Carnforth, UK), was dissolved in 0.2 N NaOH and then back titrated to pH7 with HCl. An additional group of 16 rats received subcutaneous injections of either SnPP (10 μmoles/kg of body weight) or vehicle 12 h prior to PHx, bleeding, and RBC resuscitation. They were subsequently sacrificed one day post-PHx (POD1).

#### 2.1.5. Isolation of Hepatic Monocytes by FACS Sorting

Rats were sacrificed one day post-PHx, and their livers were perfused at a flow rate of 10 mL/min using 100 mL of PPML (5.36 mM KCl, 0.44 mM KH_2_PO_4_, 4.17 mM NaHCO_3_, 138 mM NaCl, 0.38 mM Na_2_HPO_4_, 5 mM glucose, 0.5 mM EDTA, 50 mM Hepes, PH7.35) and 100 mL PM (0.78 mM MgSO_4_, 0.44 mM MgCl_2_, 0.134 mM Na_2_HPO_4_, 0.44 mM KH_2_PO_4_, 5.36 mM KCl, 50 mM Hepes, 138 mM NaCl, 11 mM dextrose, 2 mM CaCl_2_, 4% BSA, PH7.4), containing liberase (0.04 mg/mL; Roche, Indianapolis, IN, USA). The perfused liver was excised and incubated in PM/liberase solution, and immune cells were isolated by centrifugation. After perfusion and hepatocyte removal, cells were incubated with PE-conjugated anti-rat CD11b antibody (Biolegend, San Diego, CA, USA) and sorted using a FACSAria III Cell Sorter (BD Biosciences, San Jose CA, USA). Total RNA was extracted from CD11b+ cells with an RNeasy Mini Kit (Qiagen, Austin, TE, USA) and stored at −80 °C.

### 2.2. Histological Evaluation

#### 2.2.1. Histology and Immunohistochemistry

Liver samples used for histological evaluation were fixed in formalin, paraffin-embedded, and stained with hematoxylin and eosin (H&E). Apoptosis was assessed by immunostaining with TUNEL (Roche Diagnostics Corp, Indianapolis, IN, USA). Slides were counterstained with DAPI. To assess cell proliferation, BrdU (Sigma Aldrich, St Louis, MO, USA) was injected I.P. (100 mg/kg) into rats three hours prior to euthanasia, and liver tissue sections were stained as follows: antigen retrieval was performed with citrate buffer (0.01 M), and sections were incubated with mouse anti-BrdU antibody (1:200; Neomarker, Fremont, CA, USA). Hepatic macrophages were observed by antigen retrieval with protease type XXIV (Sigma) and immunostained with monoclonal mouse anti-rat CD68 (ED1; BioRad, Hercules, CA, USA) for total macrophages and monoclonal mouse anti-rat CD163 (ED2; BioRad, Hercules, CA, USA) for activated macrophages. PAI-1 was detected with rabbit anti-PAI-1 antibody (1:200; Abcam, Cambridge, UK). For each immunostaining, the number of positive cells per high power field (HPF; magnification ×200) was counted in 10 randomly selected fields per liver (6 rats/group/time point), and the mean value ± SD was determined.

#### 2.2.2. Measurement of Cell Size

Digital imaging of 5 μm liver sections prepared from formalin-fixed, paraffin-embedded samples stained with mouse anti-β-catenin (BD Biosciences, San Jose, CA, USA) following antigen retrieval with citrate buffer (0.01 M), followed by goat anti-mouse IgG Alexa 488 (1:400; Abcam) were photographed at a magnification of ×400. Hepatocyte size was measured as area in pixels using ImageJ (NIH) software on 100 hepatocytes per rat.

### 2.3. Bioinformatics Analysis

#### 2.3.1. Gene Expression Profiling

Total RNA was isolated from frozen liver tissues with TRIzol reagent according to the manufacturer’s instructions (n = 3 rats/group). Amplified and biotinylated sense-strand DNA was prepared according to the standard Affymetrix protocol from a total of 100 ng RNA (Expression Analysis WT PLUS Technical Manual 2013, Affymetrix). Following fragmentation, 2.3 μg of biotinylated sense-strand DNA was hybridized for 16 h at 45 °C on a GeneChip Rat Gene 2.0 ST Array. GeneChips were washed and stained in the Affymetrix Fluidic Station 450 and were then scanned using an Affymetrix Gene Chip Scanner 3000. Pre-processing was performed with Affymetrix tools and two algorithms for summarizing microarray probes, namely, the Robust Multichip Average (RMA) algorithm and the Probe Logarithmic Intensity Error (PLIER), which were deployed to increase the statistical power of gene differentiation analysis. Gene and gene ontology (GO) annotations were determined for each probe set according to the Affymetrix annotation files. Differentially expressed gene sets were then selected, with the *t*-test *p*-value cut-off set for accumulating reasonably sized gene sets. With these sets of genes, an enriched GO term analysis was performed, leading to GO-based clustering. We performed a KEGG-based pathway search and GO enrichment analysis on the log-normalized expression values. The data discussed in this publication have been deposited in NCBI’s Gene Expression Omnibus and are accessible through GEO Series accession number GSE216924 (https://www.ncbi.nlm.nih.gov/geo/query/acc.cgi?acc=GSE216924 (accessed on 5 September 2022)).

#### 2.3.2. qRT-PCR

Liver tissue was homogenized, and RNA was extracted with the RNeasy Mini Kit (Qiagen, Austin, TX, USA) according to the manufacturer’s instructions. cDNA synthesis was performed using the High-Capacity cDNA Reverse Transcription Kit (Applied Biosystems, Cheshire, UK). Quantitative PCR was used to determine the levels of expression by means of an SYBR Green PCR Kit (QuantaBio, Beverly, MA, USA) on the CFX384 Touch Real-Time PCR Detection System (Bio-Rad, Hercules, CA, USA). The relative expression of target genes was normalized to HPRT housekeeping gene levels. The primer sequences used are listed in Table 1.

#### 2.3.3. Protein Extraction and Western Blotting

Total protein was extracted from the livers with Radioimmuno Precipitation Assay lysis buffer. Cell lysates containing 50 μg of total protein were subjected to SDS-PAGE gels and transferred to nitrocellulose membranes (1704158; Bio-Rad, Hercules, CA, USA). The membranes were blocked in 1% non-fat milk and then incubated with the primary antibody, mouse anti-human HO-1 monoclonal IgG1 Kappa (Stress Marq), at a dilution of 1:1000 or with anti-β-actin (MP Biomedicals, Irvine, CA, USA) overnight at 4 °C. The signals were developed with an enhanced chemiluminescence solution (Bio-Rad, Hercules, CA, USA) and visualized on a Bio-Rad bioluminescence device. The intensity of the bands was quantified using ImageJ, and the results were normalized to actin.

### 2.4. In Vitro Analysis

#### Co-Cultures of Human RBCs and Macrophages

**Human monocytes** were isolated from the peripheral blood of human donors using the Human Monocyte Isolation Kit (EasySep, StemCell Technologies) and Ficoll according to the manufacturer’s instructions and seeded on 12-well plates at 5 × 10^5^/well in RPMI medium (Gibco, NY, USA) containing 10% FCS (Gibco, NY, USA), 1% penicillin/streptomycin (Biological Industries, Beit Haemek, Israel) and 1% L-glutamine (Hyclone, Logan, UT, USA).

**RBCs** were isolated from human peripheral blood as described above in *RBC collection and storage* and either stored at 4°C for 21 days (S-RBCs) or used the same day (F-RBCs). F-RBCs or S-RBCs from the same donor were added two days after seeding the peripheral blood monocytes. Conditioned media (CM) from this co-culture was collected 24 h later.

**HepaRG cells** were cultured in Williams Medium (Gibco, NY, USA) containing 10% FCS (Gibco, NY, USA), 1% penicillin/streptomycin (Biological Industries, Beit Haemek, Israel), 1% L-glutamine (Hyclone, Logan, UT, USA), 0.05 mM hydrocortisone (Sigma Aldrich, Taufkirchen, Germany) and 0.5% insulin and seeded on 96-well plates at 1 × 10^4^ cells/well for 48 h. The cells were then cultured in a starvation medium (Williams medium with 1% FCS) for 24 h prior to the addition of 200 μL CM per well for an additional day. The effect of the CM on HepaRG cell proliferation was determined by using an XTT proliferation assay (Biological Industries, Beit Haemek, Israel). To identify secreted factors in the CM, we used a Proteome Profiler Antibody Array (R&D Systems, Minneapolis, MN, USA) according to the manufacturer’s instructions.

### 2.5. Statistical Analysis

One-way analysis of variance (ANOVA) for repeated measurements was used to determine statistical significance between experimental groups, followed by the Student–Newman–Keuls post hoc test. In cases where data were distributed unevenly, the Mann–Whitney U test was used to determine statistically significant differences between groups. For multiple comparisons, the *p*-value was subsequently adjusted with the Bonferroni method. Paired Student’s *t*-test (two-tailed) was used for comparisons within groups. We considered a two-tailed value of *p* < 0.05 statistically significant. Results are presented as means ± SD. Data were analyzed using GraphPad (Prism 8.02; San Diego, CA, USA).

## 3. Results

### 3.1. The Effect of RBC Storage Time on Hepatocyte Proliferation and Hypertrophy

In order to determine the effect of stored RBCs transfusion on liver regeneration following PHx, we performed 50% PHx on Sprague-Dawley rats, bled them, administered F-RBCs or S-RBCs, and then quantified hepatocyte proliferation at various times post-PHx, by BrdU incorporation. One day post-PHx, approximately 27% and 20% of liver cells were BrdU-positive in control hepatectomized rats (PHx) and rats that were bled and transfused with F-RBCs (F-RBC), respectively (*p* > 0.05). The percentage of BrdU-positive cells in both cases was significantly higher than in rats that underwent PHx and bleeding without resuscitation, in which only 2% of hepatocytes were BrdU-positive [24]. Interestingly, following transfusion with S-RBC, only 5% of the cells were BrdU-positive, revealing similar proliferation rates to those of rats that were bled and not transfused, suggesting that administration of S-RBCs halts liver cell proliferation one day post-PHx (Figure 1A,B). Four and seven days after surgery, the number of BrdU-positive cells declined to approximately 3% in all groups, in accordance with past studies showing that in rats, peak hepatocyte proliferation occurs within 24 h of PHx [3]. Concurrently, liver mass restoration was significantly attenuated on the first four days post PHx and S-RBC resuscitation (Figure 1C).

Following PHx, significant weight loss is one of the predictors of adverse outcomes post-surgery. To determine the effect of transfusion of F-RBCs versus S-RBCs on weight loss following PHx, we recorded the body weight of the rats prior to surgery and again preceding sacrifice. As opposed to a 3.7% body weight loss in rats that received a transfusion of F-RBCs following PHx, those animals receiving S-RBCs lost 6.5% of their body weight (*p* < 0.001).

In healthy animals and humans, liver regeneration is a tightly regulated process achieved through a combination of hypertrophy and hyperplasia and involves resident liver cell populations as well as growth factors, cytokines, and extracellular matrix, in addition to other regulators. We showed previously that in rats that underwent PHx and controlled bleeding in the absence of RBC resuscitation, hepatocytes undergo hypertrophy to compensate for their inability to proliferate effectively [24]. In this study, we measured the size of hepatocytes in liver sections one day after PHx, controlled bleeding, and RBC transfusion. The cell size of hepatocytes from rats receiving S-RBCs was not significantly different from the size of hepatocytes in the livers of control naive rats. In contrast, enlarged cell size was observed in groups of animals after PHx that were not transfused as well as in animals receiving blood transfusions with F-RBCs (Figure 1D,E). This suggests that hypertrophy of hepatocytes is diminished by S-RBC transfusion and that administering F-RBCs may be beneficial to liver regeneration. It is interesting to note that the delay in liver regeneration due to resuscitation with S-RBCs was also associated with increased apoptosis (Figure 1F,G). However, the overall number of apoptotic cells was low, suggesting that this mechanism only plays a minor role in inefficient liver regeneration.

### 3.2. Blood Count and Liver Enzymes following PHx and Blood Transfusion

PHx, concomitant with bleeding and administration of S-RBCs, caused a significant reduction in RBC count, hemoglobin levels, and hematocrit one day post-PHx as compared to controls and those transfused with F-RBCs (Appendix A). However, two days after PHx, F-RBC transfusion caused a similar reduction. We further checked prothrombin time in rats, one and two days post-PHx, and resuscitation. In both F-RBC and S-RBC groups, measurements were slightly increased but stayed within normal limits (Appendix A). Resuscitation with S-RBCs was associated with increased liver injury six hours after surgery, as reflected by the significantly elevated ALT levels (2.3-fold, *p* < 0.05; Appendix A) as compared to rats post-PHx that were transfused with F-RBCs. This observed liver damage could lead to attenuation in liver regeneration.

### 3.3. Initiation and Termination of Cytokine Levels in Livers following PHx and RBC Transfusion

Cytokines are known to play key roles as mediators in both the initiation and termination phases of liver regeneration [25]. We determined the levels of initiation and termination cytokines present in liver tissues at 6 and 12 h post-PHx. Six hours post-PHx, IL-6 initiation cytokine expression in rats transfused with F-RBCs was significantly higher than in animals after PHx alone or those receiving S-RBCs. IL-6 expression 24 h post-PHx and TNF-α levels at either time point showed insignificant differences between groups (Figure 2). In contrast, HGF expression was significantly upregulated in rats receiving S-RBCs, at both time points. The termination cytokine, TGF-β, was expressed at significantly higher levels in the S-RBC group than in the animals receiving F-RBC six hours after surgery, while IL1-β mRNA levels were not significantly different between these groups at either time point (Figure 2). While IL-6 and TNF-α play a critical role in the initiation and progression of liver regeneration, IL-10 was shown to negatively regulate liver regeneration via suppression of the inflammatory response. In our samples, there was no significant difference in levels of IL-10 expression between groups at either time point. Taken together, the types of cytokines expressed following PHx in the livers of rats receiving F-RBCs versus S-RBCs are not likely to be the main regulators of the altered regeneration kinetics.

### 3.4. Storage Time of Transfused RBCs Affects Expression of Genes Associated with Tissue Regeneration

In order to determine genes and pathways involved in differing liver regeneration rates that depend on the storage time of transfused RBCs, we performed transcriptome profiling (RNA-seq) on total RNA extracted from the livers of 12 rats, one day post-PHx. Bioinformatics analysis revealed numerous genes that were differentially expressed by more than two-fold between the different groups of rats. Principle component analysis (PCA) of the liver samples for all probe sets available in the Affymetrix array using a covariance matrix shows that the three replicates of each group cluster together, and the overall expression levels of each group are distinctly different from other groups (Figure 3A). GO enrichment analysis showed large differences in particular molecular pathways between rats transfused with F-RBCs and those receiving S-RBCs (Appendix A). We further checked for enriched molecular pathways based on the RMA preprocessing algorithm. Figure 3 shows heat maps of four selected pathways that significantly differ between groups of rats receiving F-RBC and S-RBC resuscitation: cell cycle (Figure 3B), DNA replication (Figure 3C), platelet activation (Figure 3D), and chemokine signaling (Figure 3E). The top 30 genes for each pathway are listed (Appendix A).

### 3.5. Expedited Macrophage Activation following Transfusion with S-RBCs

Studies have shown that macrophages play a major role in the initiation of liver regeneration. Therefore, we decided to examine the role of macrophages in the process of liver regeneration following PHx combined with RBCs transfusion. Initially, we quantitated the number of macrophages that were present in rat livers post-surgery by immunostaining liver tissue sections with an antibody against the macrophage marker CD68 (ED1). During the first two days post-PHx, the number of CD68+ cells increased; however, no significant difference was detected in the percentage of macrophages between the livers of control animals (PHx) and those receiving transfusions of F-RBCs or S-RBCs, post-PHx (Figure 4A,B). Yet, four days after PHx, the percentage of CD68+ cells decreased 8-fold in rats transfused with S-RBCs as compared with either untranfused rats post-PHx or those receiving F-RBCs (*p* < 0.001). The difference in the percentage of CD68+ cells was resolved seven days post-PHx, and livers in all groups contained a similar number of CD68+ cells. We then stained liver tissue sections with a marker of activated macrophages using an anti-CD163 (ED2) antibody. One day after PHx, significantly more CD163+ cells were observed in rats transfused with S-RBCs as compared to untransfused rats post-PHx and rats administered F-RBC transfusion (*p* = 0.02; Figure 4C,D). However, this expedited response was short-lived, and by four days post-PHx, the percentage of CD163+ cells in this group was 4-fold less than the control and F-RBC-transfused groups. As with CD68+ cells, the percentages of CD163+ cells were similar in all groups seven days post-PHx. These results indicate that the activation of macrophages immediately following PHx, perhaps caused by the presence of S-RBCs, may prematurely halt liver regeneration.

Intracellular adhesion molecule (ICAM)-1, a member of the immunoglobulin superfamily, is a well-described adhesion molecule expressed on many cell types, including endothelial cells, epithelial cells, and fibroblasts. ICAM-1 is overexpressed in response to pro-inflammatory mediators such as TNF-α and IL-1. By using ICAM-1^−/−^ mice, the central role of ICAM-1 and leukocyte recruitment in triggering liver regeneration in vivo was demonstrated [29]. Since we noticed increased macrophage activation in rats administered S-RBCs, we assessed the expression of ICAM-1 in rat liver samples obtained one day post-PHx. Indeed, a significantly higher percentage of blood vessel cells were positive for ICAM-1 in rats that received S-RBC compared to rats receiving F-RBC (Figure 4E,F). The elevated ICAM-1 level may contribute to macrophage activation.

### 3.6. The Role of Macrophages in Liver Regeneration following PHx

The next step was to ascertain the role of macrophages in our in vivo model by monitoring hepatocyte proliferation after depleting the hepatic macrophage cell population prior to PHx. Towards this end, liposomal clodronate, a potent anti-macrophage agent, was administered to the rats 24 h prior to PHx, and we measured hepatocyte proliferation by BrdU incorporation. In agreement with previous reports, we observed the absence of macrophages upon liposomal clodronate administration. Interestingly, one-day post-PHx, hepatocyte proliferation increased in all animals that underwent PHx following macrophage depletion as compared to the non-depleted groups, with a significant elevation in rats that were transfused with S-RBCs (Figure 4G,H). Taken together, it is tempting to suggest that activated hepatic macrophages may suppress liver regeneration following PHx and that this inhibitory effect may be enhanced by transfusion with S-RBCs.

### 3.7. The Effect of Conditioned Media (CM) from Macrophage-RBC Co-Cultures on Hepatocyte Proliferation

To further elucidate the underlying mechanism by which macrophages affect liver regeneration after PHx and blood transfusion, we established an in vitro system. Monocytes isolated from five healthy human donors were co-cultured for 24 h with either F-RBCs or RBCs drawn from the same donor and stored for 21 days (S-RBC). CM from the co-culture was collected and added to the HepaRG human hepatoma cell line. After 24 h, HepaRG proliferative ability was analyzed using the XTT proliferation assay. The proliferation of cells grown in CM from the macrophages + S-RBC co-culture was significantly inhibited in contrast to proliferation levels of cells grown in CM from F-RBCs only or macrophages co-cultured with F-RBCs (Figure 5A). These results infer a possible macrophage-mediated inhibition of hepatocyte proliferation.

To broaden our understanding of the pathways involved in this inhibitory effect, we performed a Proteome Profiler Antibody Array on the CM described above (Figure 5B). This assay allowed us to identify the factors that are secreted into the co-culture medium that may affect hepatocyte proliferation. Few pro-inflammatory cytokines, including IL-17A and IL-8, were expressed at higher levels in the CM from the macrophage + S-RBC co-culture than in other conditions (Figure 5B,C). In contrast, angiogenin, which is known to reduce immune inflammation in certain contexts [30], was suppressed in the macrophage + S-RBC co-culture CM. Our data hint at the possibility that pro-inflammatory cytokines secreted from macrophages exposed to S-RBCs are an additional process by which hepatocyte proliferation is inhibited [31].

### 3.8. The Function of Heme Oxygenase-1 (HO-1) in Liver Regeneration

One of the genes that were significantly elevated in the livers of rats that were subjected to PHx, bleeding, and resuscitation with S-RBC is HO-1. Recently, several studies dispute the classic M1/M2 macrophage theory, claiming that additional subgroups of macrophages exist [32]. Among these new subgroups are the hemorrhage-activated macrophages, Mhem, which are responsible for depleting the blood of old RBCs by phagocytosis of erythrocyte remnants and hemoglobin deposits. Mhem macrophages are distinguished from other macrophage subgroups by their elevated HO-1 activity. To verify the role of activated Mhem macrophages in halting regeneration following PHx and S-RBC transfusion, we measured levels of HO-1 mRNA and protein in the livers of the rats in our in vivo model. We observed significantly elevated HO-1 mRNA levels in livers from PHx rats that were transfused with S-RBCs as compared to those receiving F-RBCs, 24-h post-PHx (Figure 6A,B). Likewise, Western blot analysis of HO-1 expression, at this same time point post-PHx, showed higher levels of this protein in the livers of the S-RBC group than in the F-RBC group (Figure 6C,D). As heme oxygenase is known to convert heme to bilirubin and free iron, we assessed the existence of iron deposits in the livers of rats that received S-RBCs, by Perl’s Prussian blue staining. Only rare spots of iron deposits were identified in all rats (data not shown), negating the possibility of iron toxicity as a cause for delayed liver regeneration. 

To further validate the inhibitory role of HO-1 on liver regeneration, we administered tin protoporphyrin IX (SnPP), a known HO-1 inhibitor, four hours prior to PHx and measured hepatocyte proliferation rates as detected by BrDU incorporation (Figure 6E,F). In rats receiving F-RBC transfusions, the number of BrdU+ cells was slightly increased upon HO-1 inhibition, although the difference was not significant. Interestingly, HO-1 inhibition resulted in significantly increased numbers of BrdU+ cells in rats that were transfused with S-RBCs (Figure 6E,F). Furthermore, when assessing liver damage, SnPP administration lowered levels of ALT in the serum of the animals receiving S-RBCs post-PHx (170 ± 15 compared to 270 ± 25, respectively; *p* = 0.01). Together, we show that activated macrophages play a detrimental role in liver regeneration, particularly following the transfusion of S-RBCs. It is likely that Mhem macrophages are key players in this process due to their expression of HO-1.

### 3.9. Elevated PAI-1 Expression following Transfusion with S-RBCs

RNA-seq results from isolated monocytes revealed an elevation of genes associated with plasmin activation in macrophages isolated from the livers of rats that were subjected to PHx, bleeding, and resuscitation with S-RBC. As plasminogen activator inhibitor-1 (PAI-1) was previously suggested to adversely affect liver regeneration, we measured protein levels in the liver six hours post-PHx, using immunostaining. We found a three-fold increase in PAI-1 levels in rats that received S-RBCs compared to all other groups (Figure 6G,H). Our results implicate activated macrophages in delayed liver regeneration following S-RBC transfusion via both HO-1 and PAI-1 overexpression.

## 4. Discussion

Extensive liver resections are common in patients suffering from severe liver damage and in patients with liver tumors. Liver regeneration plays a critical role in the recovery of patients who have undergone PHx for the removal of a primary tumor or hepatic metastasis or in the case of living-donor liver transplantation. Blood loss frequently occurs during these operations, thereby resulting in patients receiving blood transfusions [33,34]. Since delayed liver regeneration sometimes leads to liver failure, we were interested in understanding the role of blood transfusions on the rate of liver regeneration. In particular, we focused on elucidating the effect of RBC storage time on liver regeneration. In past studies, we assessed the impact of acute bleeding on hepatocyte proliferation and cytokine production after PHx [24]. In the current study, we attempted to uncover the effect of transfusion with F-RBCs versus S-RBCs in a rat model while exploring possible contributing molecular mechanisms. The deleterious effects of RBC storage prior to transfusion have been previously documented and include post-transfusion enhanced clearance, plasma transferrin saturation, nitric oxide scavenging, and immunomodulation [35]. However, the effect of storage time in the context of PHx and liver regeneration is unknown. Herein, we show that transfusion of F-RBCs following PHx and controlled bleeding restores the natural ability of the liver to regenerate quickly and efficiently. Conversely, blood transfusions with S-RBCs inhibit and slow the regeneration process. The clinical ramifications of these results might indicate that more focus should be placed on transfusing newly collected blood to patients undergoing liver resection. 

Several studies have emphasized the role of immune cell populations in liver regeneration, demonstrating complex interactions between KCs, endothelial cells, and hepatocytes by direct contact or via the production of cytokines. Macrophages also play an important role in the regulation of liver homeostasis in physiological conditions and in pathology [36]. Previously, it was suggested that activation of KCs is necessary for the optimal regenerative ability of the liver, possibly through the release of TNF-α and IL-6, which initiate hepatocyte proliferation in vivo [25]. Moreover, several studies have shown that the depletion of resident liver macrophages inhibits regeneration, suggesting the possibility that macrophages play a supportive role in liver regeneration [26,37]. In contrast to these studies, we observed improved liver regeneration after macrophage depletion, as witnessed by increased hepatocyte proliferation over controls, particularly in the days immediately following PHx, suggesting that these immune cells directly influence the regeneration process. The disparity in results between past studies and the present study may be explained by differences in experimental conditions. In this study, the animals underwent controlled bleeding and RBC transfusion in addition to PHx to better mimic the clinical setting. The bleeding and, specifically, transfusion with S-RBCs may cause increased macrophage activation, already on day one post-PHx, possibly having an inhibitory effect on the regeneration process.

Macrophages constitute an extremely heterogeneous population, which has historically been divided into two main classes: M1 and M2. M1 macrophages are characterized by the expression of high levels of pro-inflammatory cytokines and strong antimicrobial activity. In contrast, M2 macrophages function in resolving inflammation while promoting cell proliferation and wound healing. Although significantly more activated macrophages were detected in the livers of rats that received S-RBC, these cells did not express high levels of arginase (data not shown) and IL-10, suggesting that they are not classical M2 monocytes. Recently, several novel macrophage subsets have been identified. These subsets include hemorrhage-specialist macrophages (Mhem). Mhem are characterized by increased expression of IL-10 and CD163, which is a scavenge receptor for the Hb-haptoglobin (Hp) complex. The induction of this macrophage population was also accompanied by the upregulation of HO-1, the rate-limiting enzyme that catalyzes heme degradation. HO-1 catabolizes heme to carbon monoxide (CO), ferrous iron, and biliverdin, which is converted immediately to bilirubin [38]. These byproducts of HO-1 enzymatic activity are regarded as cytoprotective molecules due to their antioxidant and anti-inflammatory properties [39]. Moreover, these products were shown to be anti-proliferative [40]. Recent data suggest a key role for HO-1 in the regulation of cellular homeostasis and growth [40], although its effects are cell-type specific and can have the opposite result. In the context of liver regeneration, a previous study demonstrated that inhibition of HO-1 enhances rat liver regeneration after PHx, concomitantly with a rapid increase in the levels of inflammatory mediators such as IL-6, phospho-JNK, and phospho-STAT3, suggesting that HO-1 acts as a negative modulator of liver regeneration [40]. Our results also demonstrate that elevated HO-1 hepatic levels at 24 h post-PHx inhibit liver regeneration.

PAI-1 is synthesized by numerous types of cells, including vascular endothelial cells, macrophages, and fibroblasts. It acts as a potent inhibitor of fibrinolysis by regulating urokinase plasminogen activator (uPA), tissue plasminogen activator (tPA), plasmin, and matrix metalloproteinase (MMP) proteolytic activity, as well as fibrin levels. Several studies demonstrated that uPA is one of the factors that is involved in the initiation of liver regeneration by activating metalloproteinase and HGF release [41,42]. PAI-1 is the main inhibitor of uPA. It was previously shown that PAI-1 causes liver injury following bleeding and blood resuscitation. Moreover, the administration of the HO-1 inhibitor, SnPP, also inhibits PAI-1 expression [43]. Therefore, our results showing the elevation of PAI-1 in the livers of rats that were resuscitated with S-RBC may infer an additional mode of inhibition of the regeneration process, specifically due to RBC storage time.

The process of liver regeneration contains remarkable redundancy between signals. Many of the signaling pathways overlap in function, leading to complete, albeit delayed, regeneration. Our data reveal that S-RBC resuscitation has a harmful effect on liver regeneration in rats following PHx and bleeding. We show that activated macrophages play an adversative role in liver regeneration. However, as macrophage depletion is a drastic therapeutic approach, we searched for other, more clinically relevant ways to block the detrimental effects of activated macrophages without resorting to depletion. In light of our data showing elevated levels of HO-1, which principally characterizes Mhem macrophages [44], we administered an HO-1 inhibitor prior to PHx. Histological findings showed that regeneration was significantly enhanced in animals after HO-1 inhibition, regardless of RBC storage time. The success of the HO-1 inhibitor in restoring liver regeneration improves our understanding of the possible mechanisms blocking liver regeneration following PHx and lends hope to find an effective and safe way to ameliorate the detrimental effects of blood loss and stored blood transfusion on patients undergoing liver resection. As liver surgery is frequently performed in patients with a pre-existing damaged liver, such as in patients with liver fibrosis/cirrhosis, further studies should characterize the effect of bleeding and transfusion on these conditions as well.

## Figures and Tables

**Figure 1 cells-11-03522-f001:**
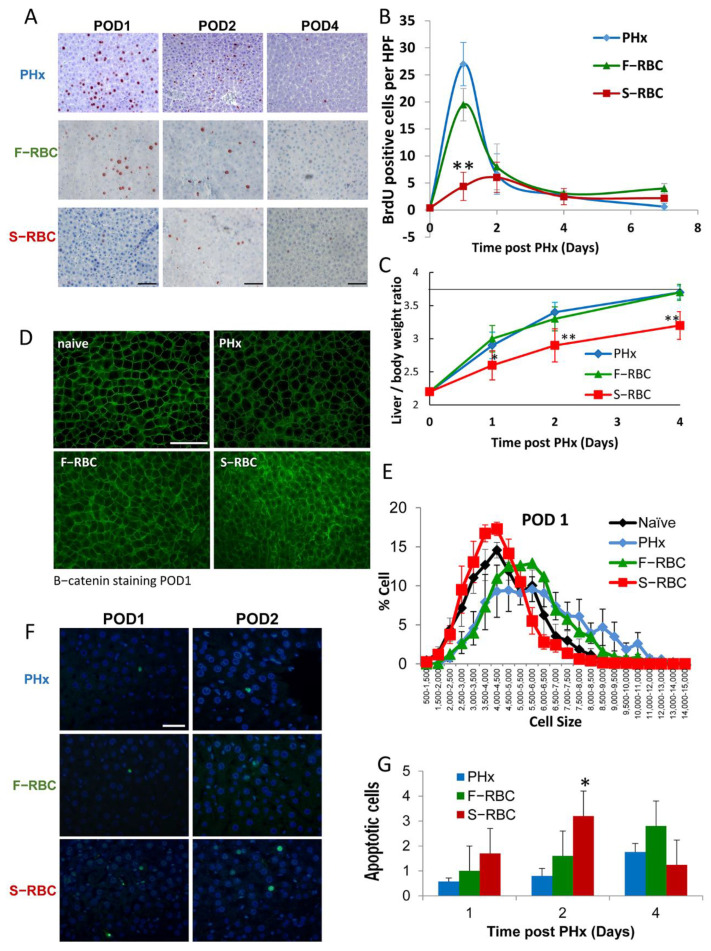
Transfusion with S-RBC following PHx and bleeding inhibits hepatocyte hyperplasia and hypertrophy. (**A**) Representative images of BrdU staining showing hepatocyte proliferation in the various groups 1, 2, and 4 days following PHx (POD = post-operative day); 400× magnification. Bar = 100 μm and apply for all. (**B**) Quantification of the number of BrdU-stained cells per HPF. Average of 10 fields per animal from the different liver lobes. n = 5–10 rats per group. (**C**) Liver to body weight ratio for the various groups 1, 2, and 4 days following PHx. (**D**) Representative images of β-catenin cell membrane staining of hepatocytes. Bar = 100 μm and apply for all. (**E**) Hepatocyte cell size distribution in the various groups on POD1; 400× magnification. n = 3–5 per group. (**F**) Representative images of green fluorescent-labelled apoptotic cells with DAPI staining of the nuclei in the different groups, Bar = 50 μm and apply for all. (**G**) Quantitation of TUNEL-positive cells counted in 10 high-power fields/slide. Data represent mean ± SD (n = 6 rats/time point). * *p* < 0.05, ** *p* < 0.01.

**Figure 2 cells-11-03522-f002:**
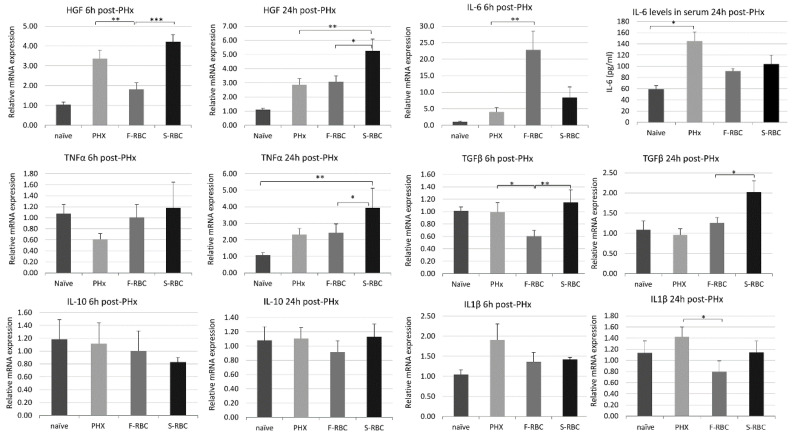
Hepatic cytokine levels after PHx are altered by RBC transfusion. Initiation and termination cytokine levels expressed in livers of the various experimental groups, 6 and 24 h post-PHx. RNA levels were measured by RT-PCR, except than IL6 protein level at 24 h, which was measured by ELISA on serum samples. * *p* < 0.05, ** *p* < 0.01, *** *p* < 0.001.

**Figure 3 cells-11-03522-f003:**
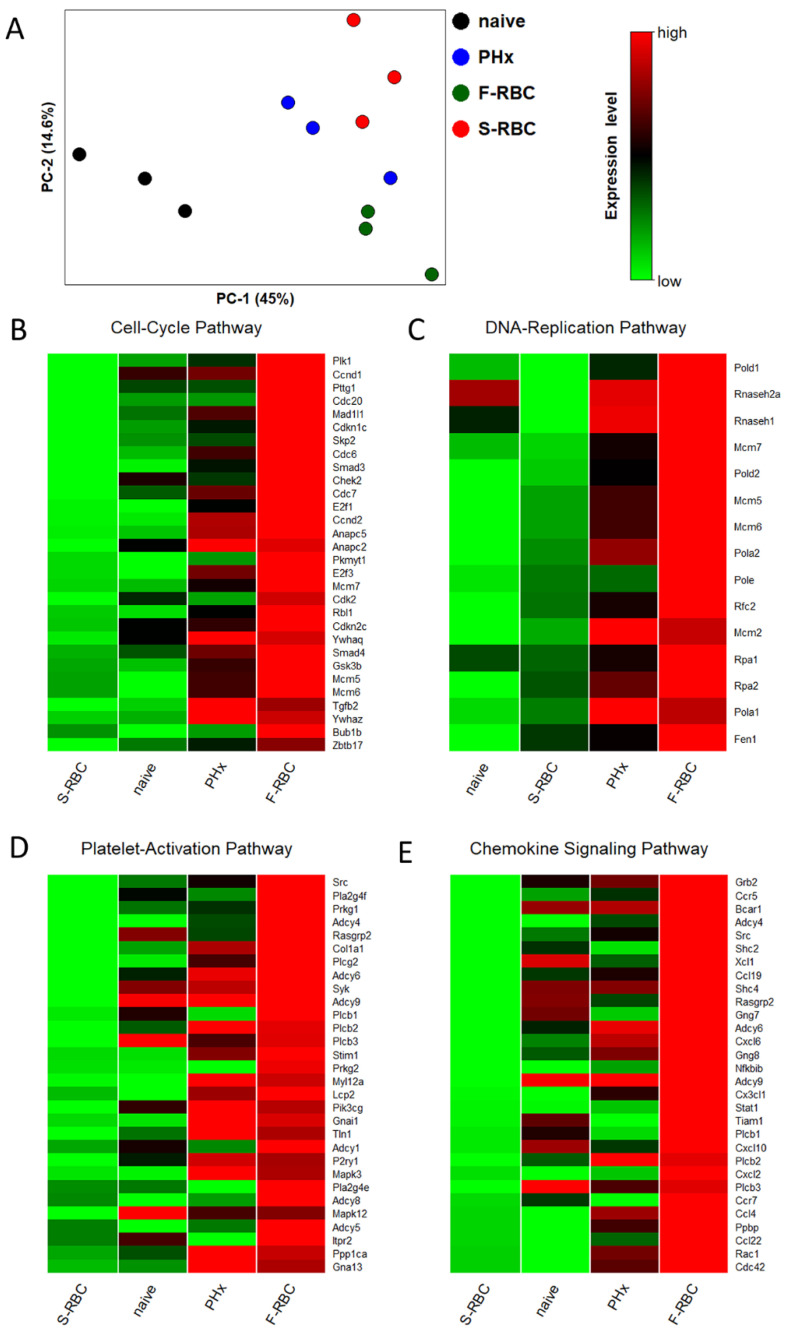
Effect of F-RBC vs. S-RBC resuscitation on cell cycle and inflammatory pathway genes. (**A**) Principal Component Analysis (PCA) plot showing RNA-Seq samples analyzed by treatment groups. PHx substantially alters gene expression, as demonstrated by PC-1. PCA analysis confirms that the three samples of each group cluster together and that there is a significant difference between the RBC-resuscitated groups. (**B**–**E**) Significant differences are observed between resuscitation with F-RBCs as compared to S-RBC in several pathways. The top 30 differentially expressed genes between groups receiving F-RBC resuscitation and S-RBC resuscitation are visualized as heat maps of gene expression, where red indicates upregulated genes, and green indicates downregulated genes.

**Figure 4 cells-11-03522-f004:**
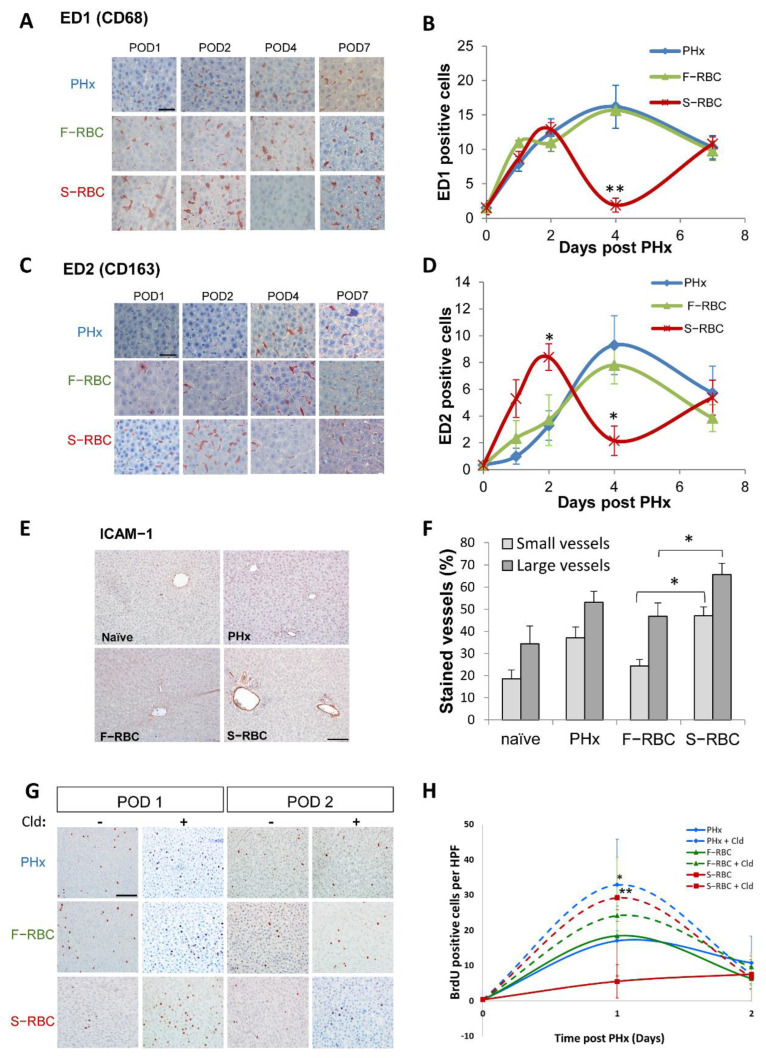
Transfusion with S-RBC following PHx and bleeding causes changes in the number of hepatic macrophages and activated macrophages. (**A**) Representative images of liver sections stained with anti-CD68, showing macrophages in the liver in the various groups on days 1, 2, 4, and 7 following PHx (POD = post-operative day); Bar = 50 μm and apply for all. (**B**) Quantification of the number of CD68+ cells per HPF. Average of 10 fields per animal from the different liver lobes. n = 5–10 per group. (**C**) Representative images of liver tissue sections stained with anti-CD163 showing activated macrophages in the liver in the various groups, 1, 2, 4, and 7 days following PHx (POD = post-operative day); Bar = 50 μm and apply for all. (**D**) Quantification of the number of CD163+ cells per HPF. Average of 10 fields per animal from the different liver lobes. N = 5–10 per group. (**E**) Representative images of ICAM-1 staining, one day post-PHx; Bar = 100 μm. (**F**) Quantification of the percent of ICAM-1-stained vessels per HPF. Average of 10 fields per animal from the different liver lobes were analyzed. N = 5–10 per group. (**G**) Representative images of BrdU staining showing hepatocyte proliferation in the various groups of rats on days one and two following PHx with (+) or without (−) macrophage depletion by Cld administration (POD = post-operative day); Bar = 100 μm. (**H**) Quantification of the number of BrdU-positive cells per HPF. Average of 10 fields per animal from the different liver lobes. n = 5–10 per group. * *p* < 0.05, ** *p* < 0.01.

**Figure 5 cells-11-03522-f005:**
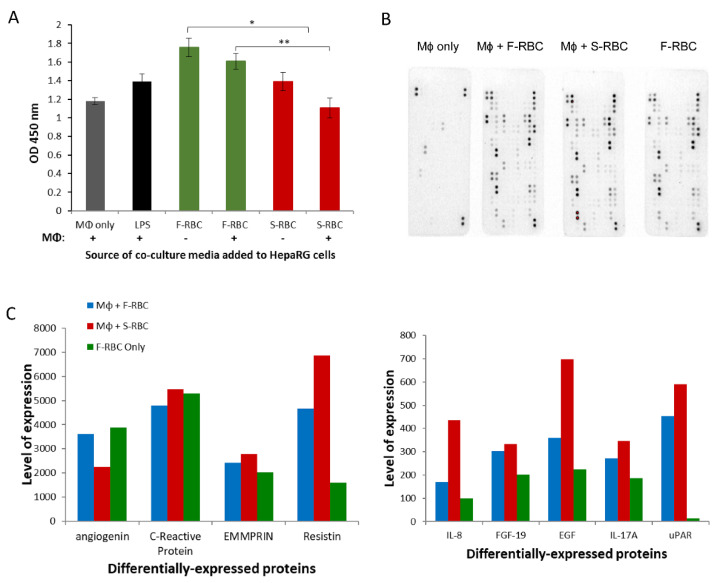
Conditioned media (CM) from macrophage-RBC co-cultures influences hepatocyte proliferation in-vitro. (**A**) HepaRG proliferation after addition of CM from F-RBCs or S-RBCs co-cultured in the presence (+) or absence (−) of macrophages (MΦ), macrophages alone, and macrophages activated with LPS. (**B**) Images of Proteome Profiler Antibody Array membranes corresponding to the different CM collected. (**C**) Differentially expressed proteins in CM from macrophages + F-RBC co-culture, macrophages + S-RBC co-culture, and F-RBCs only. * *p* < 0.05, ** *p* < 0.01.

**Figure 6 cells-11-03522-f006:**
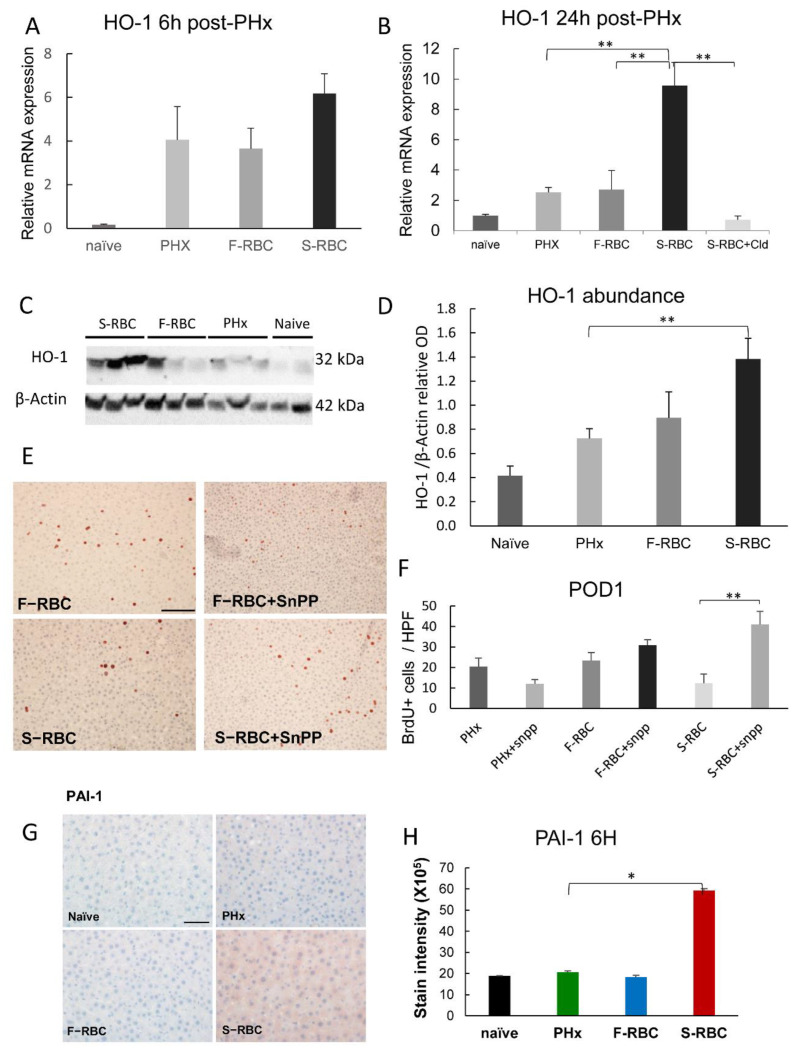
HO-1 upregulation in livers of rats following PHx and S-RBC transfusion. Hepatic HO-1 expression in the various groups at 6 h (**A**) or 24 h (**B**) post-PHx. (**C**) Western blot of HO-1 in the livers of the different groups following PHx. (**D**) Western blot quantification. (**E**) Representative images of BrdU staining showing hepatocyte proliferation in the various groups one day after PHx; Bar = 100 μm and apply for all. (**F**) Quantification of the number of BrdU-positive cells per HPF. Snpp, an HO-1 inhibitor, was injected subcutaneously four hours prior to PHx. Average of 9 fields per animal from the different liver lobes. n = 5–15 per group. (**G**) Representative images of PAI-1 hepatic staining in the various groups from samples obtained six hours post-PHx; Bar = 50 μm and apply for all. (**H**) Quantification of staining intensity. Average of 10 fields per animal, n = 6 rats per group. * *p* < 0.05, ** *p* < 0.01.

**Table 1 cells-11-03522-t001:** Primers used for RT-qPCR analysis.

Target	Forward	Reverse
HGF	GATCAGGACCTTGTGAGGGA	ATGGCACATCCACGACCA
IL-6	CTTCCAAACTGGATATAACCAGGAA	CTTCACAAACTCCAGGTAGAAACG
TNFα	CCCAGACCCTCACACTCAGATC	CTCCGCTTGGTGGTTTGCTA
TGFβ	GATACGCCTGAGTGGCTGTCT	CGAAAGCCCTGTATTCCGTCT
IL1β	GCTGTGGCAGCTACCTATGTCTT	GTCACAGAGGACGGGCTCTTC
IL10	GATACAGCTGCGACGCTGTCA	CCTTGTAGACACCTTTGTCTTGGA
HO-1	GTCCAGGGAAGGCTTTAAGCT	GGCATAGACTGGGTTCTGCTT
HPRT	GAGCACTTCAGGGATTTGAATCAT	GTAGATTCAACTTGCCGCTGTCT

## Data Availability

The data presented in this study are available on request from the corresponding author.

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
