# Peer review of "Premature Macrophage Activation by Stored Red Blood Cell Transfusion Halts Liver Regeneration Post-Partial Hepatectomy in Rats"

_cells, 2022, doi:10.3390/cells11213522_

Round 1
Reviewer 1 Report
Review of the manuscript entitle: “Premature macrophage activation by stored red blood cells transfusion halts liver regeneration post partial hepatectomy in rats”
In this manuscript, the authors aimed to investigate the impact of blood storage on rat liver regeneration after partial hepatectomy followed by a controlled bleeding and blood transfusion. Thus, they could mimic the situation where a patient had an operative blood loss during a liver resection and received a blood transfusion; blood which can be stored up to 21 days.
Overall, the study is interesting and investigates a critical point in blood transfusion process in the context of liver resection. Experimental groups and timing of investigation are appropriate. Nevertheless, the manuscript flow and data organization should be improved to clarify the message and ameliorate the reading.
Please, find here some suggestions and recommendations:
· Please, show results in figure in the same order than mentioned in the text.
For example:
· PAI-1 stainings are shown in figure 3g, h but are not mentioned in the text before the very end of the manuscript, after figure 6.
· Figure 5 is mentioned right after figure 3.
· Please, use the same names for the experimental groups along the manuscript, in the figures and in the text (only naïve, PHx, PHX + F-RBCs and PHX + S-RBCs). Please, rename accordingly “OB” and “OB+SnPP” in figure 6. Please, rename accordingly “stored” and “fresh” in figure 5.
· Please, do not show data that are not discussed in the results and not described in the method section:
o Groups HEAS or bleeding in figure 5.
o Groups S-RBCs + Cld in figure 2.
Please, remove those groups from the figures or describe the method and mention them in the result section when appropriate.
In figure 6d, please discuss discordance between HO-1 mRNA and protein quantification regarding the condition S-RBC + Cld; especially because HO-1 is mainly expressed by macrophages cells in the liver.It is quite surprising to see a higher HO-1 protein content in S-RBC + Cld than in PHx condition.
Other comments:
Figure 3f: Please show SD or SEM, and mention or show conditions that are significantly different. (I.e., * significantly different from naïve)
Figure 4a: Please align the bar for statistical test comparison
Figure 4b: Please show statistical tests before concluding on this result. Also, two different scales may help the read smallest values.
Figure 5a, b, c, d: Those panels show enriched GO terms that have been selected manually. Showing an extended list as a supplemental material may provide more information to the reader. Also, showing a full list of differentially expressed genes between the four groups studied in this work could be a plus.
Figure 6c: WB traces for S-RBC +Cld are not shown here, but the corresponding quantification is shown in figure 6d.
Figure 6d: which conditions are significantly different here? Please align the line for significance to the bars.
Figure 6e: although no difference was found between the groups, please mention the experimental group corresponding to the image shown.
Author Response
- Overall, the study is interesting and investigates a critical point in blood transfusion process in the context of liver resection. Experimental groups and timing of investigation are appropriate. Nevertheless, the manuscript flow and data organization should be improved to clarify the message and ameliorate the reading. Show results in figure in the same order than mentioned in the text
• We appreciate this key critique. Accordingly, we have changed the order of the figures as follows:
- Figure 5 from the previous version is Figure 3 in the current version.
- Figure 3 from the previous version is currently Figure 4; and in addition, we removed PAI results that were shown in Figure 3G-H and instead we inserted data regarding macrophages depletion to this figure.
- Old Figure 4 is now Figure 5 and the current figure shows only the results related to the in-vitro experiments.
Please, use the same names for the experimental groups along the manuscript.
We rewritten the result section to clarify groups’ names and changed the figures accordingly.
Please, do not show data that are not discussed in the results and not described in the method section.
• Results regarding HEAS resuscitation and bleeding were removed from figure 3 (previously Figure 5). We also removed cytokine results regarding Cld depleted rats from Figure 2.
Other comments:
- Figure 3f: Please show SD- Done (in Figure 4F).
- Figure 4a: Please align the bar for statistical test comparison-
- Figure 4b: Please show statistical tests before concluding on this result. Also, two different scales may help the read smallest values. We separated the graph into two graphs with different scales. Regarding statistics – we only performed the Proteome Profiler Antibody Array assay once with a pool of CM from all five samples thus we cannot conclude regarding the statistical comparison.
- Figure 5a, b, c, d: Those panels show enriched GO terms that have been selected manually. Showing an extended list as a supplemental material may provide more information to the reader. Also, showing a full list of differentially expressed genes between the four groups studied in this work could be a plus. We added this information in supplemental table 1.
- Figure 6c: WB traces for S-RBC +Cld are not shown here, but the corresponding quantification is shown in figure 6d. We removed the quantification from the graph.
- Figure 6d: which conditions are significantly different here? Please align the line for significance to the bars. Done
- Figure 6e: although no difference was found between the groups, please mention the experimental group corresponding to the image shown. Removed from the figure.
Reviewer 2 Report
Reviewer comments
The manuscript evaluates the differences between fresh and stored blood transfusion after partial hepatectomy (PHx) in rats. Data is interesting, but the results writing is not in the sequence of the figures. Reorganize the writing with the figures. The discussion section is coherent and focused. Literature citations seem adequate. I think this study is interesting, however, there are major issues that need to be addressed before recommending acceptance.
Material and methods
Lines 93-113. I suggest to move all this information to 2.1. Animal experiments subsection.
Lines 113-118: I think is better to put this information in another subsection. There is no relationship with surgical procedure.
The authors need to add a subsection about Western Blotting with all procedures and primary and secondary antibody brand, catalog number and dilution.
Results
Lines 281-297. These results are not in order with Figure sequence. Reorganize the sequence of the results with the figures.
Lines 400-408. PAI-1 results suppose to be together with 3.5 subsection. Makes no sense put this result in the end of results since the images of this results is in figure 3.
Figure 1C and 1E. Please add a negative control for β-catenin and TUNEL to confirm the staining is specific and also add DAPI for nuclei.
Figure 5. This figure is difficult to understand. The authors use these groups for the other figures: PHx, PHx+F-RGB and PHx+S-RGB, but in Figure 5 they use other groups. Which one is PHx+S-RGB? Which one is PHx+F-RGB? This data is confused, please reanalyze.
Figure 6E. The authors didn’t show any differences in iron deposition among groups. I suggest to remove this result or put in Supplemental data or data not shown.
Author Response
The manuscript evaluates the differences between fresh and stored blood transfusion after partial hepatectomy (PHx) in rats. Data is interesting, but the results writing is not in the sequence of the figures. Reorganize the writing with the figures. The discussion section is coherent and focused. Literature citations seem adequate. I think this study is interesting, however, there are major issues that need to be addressed before recommending acceptance.
- As suggested by the reviewer, we rearranged the materials and methods section. We also add a subsection about Western Blotting with all procedures and primary and secondary antibody brand, catalog number and dilution.
- Regarding the Results section-
Lines 281-297. These results are not in order with Figure sequence. Reorganize the sequence of the results with the figures. We changed Figure order accordingly and Figure 5 is now Figure 3.
Lines 400-408. PAI-1 results supposed to be together with 3.5 subsection. Makes no sense put this result in figure We moved the results regarding PAI-1 from Fig 3 to figure 6.
Figure 1C and 1E. Please add DAPI nuclei staining in TUNEL Results to confirm the specificity.- Done
Figure 5. This figure is difficult to understand. The authors use these groups for the other figures: PHx, PHx+F-RGB and PHx+S-RGB, but in Figure 5 they use other groups. Which one is PHx+S-RGB? Which one is PHx+F-RGB? This data is confused, please reanalyze. The figure was changed based on this critique, now Fig 3.
Figure 6E. The authors didn’t show any differences in iron deposition among groups. I suggest to remove this result or put in Supplemental data or data not shown. Removed.
Round 2
Reviewer 1 Report
The authors made substantial modifications in order to address reviewer’s comments. The reviewed data organization and presentation dramatically simplify the reading of the paper and made the message clearer.
Author Response
Thank you for assisting us to improve the manuscript.
Reviewer 2 Report
I don`t have any other comments.
Author Response
We would lie to thank to the reviewers.